# Microbial Community Colonization Process Unveiled through eDNA-PFU Technology in Mesocosm Ecosystems

**DOI:** 10.3390/microorganisms11102498

**Published:** 2023-10-05

**Authors:** Siyu Gu, Peng Zhang, Shuai Luo, Kai Chen, Chuanqi Jiang, Jie Xiong, Wei Miao

**Affiliations:** 1Key Laboratory of Aquatic Biodiversity and Conservation, Institute of Hydrobiology, Chinese Academy of Sciences, Wuhan 430072, China; gusiyu@ihb.ac.cn (S.G.); pengzhang09@126.com (P.Z.); zxcluoshuai@163.com (S.L.); chenkai@ihb.ac.cn (K.C.); jiangchuanqi@ihb.ac.cn (C.J.); 2University of Chinese Academy of Sciences, Beijing 100049, China; 3School of Ecology and Environment, Tibet University, Lhasa 850000, China; 4College of Life Science and Technology, Huazhong Agricultural University, Wuhan 430070, China; 5CAS Center for Excellence in Animal Evolution and Genetics, Kunming 650223, China; 6State Key Laboratory of Freshwater Ecology and Biotechnology of China, Wuhan 430072, China

**Keywords:** metagenome, eDNA-PFU, colonization process, mesocosm, microbial communities

## Abstract

Microbial communities are essential components of aquatic ecosystems and are widely employed for the detection, protection, and restoration of water ecosystems. The polyurethane foam unit (PFU) method, an effective and widely used environmental monitoring technique, has been improved with the eDNA-PFU method, offering efficiency, rapidity, and standardization advantages. This research aimed to explore the colonization process of microbial communities within PFUs using eDNA-PFU technology. To achieve this, we conducted ten-day monitoring and sequencing of microbial communities within PFUs in a stable and controlled artificial aquatic ecosystem, comparing them with water environmental samples (eDNA samples). Results showed 1065 genera in eDNA-PFU and 1059 in eDNA, with eDNA-PFU detecting 99.95% of eDNA-identified species. Additionally, the diversity indices of bacteria and eukaryotes in both methods showed similar trends over time in the colonization process; however, relative abundance differed. We further analyzed the colonization dynamics of microbes in eDNA-PFU and identified four clusters with varying colonization speeds. Notably, we found differences in colonization rates between bacteria and eukaryotes. Furthermore, the Molecular Ecological Networks (MEN) showed that the network in eDNA-PFU was more modular, forming a unique microbial community differentiated from the aquatic environment. In conclusion, this study, using eDNA-PFU, comprehensively explored microbial colonization and interrelationships in a controlled mesocosm system, providing foundational data and reference standards for its application in aquatic ecosystem monitoring and beyond.

## 1. Introduction

Aquatic ecosystems are recognized as highly delicate and intricate systems on Earth, supporting a substantial source of the planet’s biological diversity and playing a pivotal role in upholding the ecological equilibrium of the planet [1,2,3]. Given the persistent escalation of human activities worldwide and the resulting deterioration of the ecological environment, the imperative to promptly identify, safeguard, and rehabilitate aquatic ecosystems has grown significantly [4]. The polyurethane foam unit (PFU) method was first proposed by Cairns in 1969 as a method for monitoring microbial communities in aquatic environments [5]. Compared to traditional water sampling, PFU can collect over 85% of the microbes present in the water while avoiding the issues associated with transient and random sampling in flowing water habitats such as rivers [6]. The PFU method has been employed by numerous scholars to assess the quality of both fresh and marine water in various countries, and it has consistently demonstrated its effectiveness as a rapid, economical, and accurate monitoring technique [7,8,9,10,11,12].

However, the conventional PFU method suffers from drawbacks such as a high requirement for specialized knowledge, difficulty in standardization and automation, and being time-consuming and labor-intensive [13,14]. These limitations have hindered comprehensive monitoring of microbial communities. Fortunately, the emergence of environmental DNA (eDNA) technology provides a simpler and more accurate way for microbial monitoring. The eDNA method relies on widely standardized reference sequence databases for monitoring, resulting in more accurate, objective, and comparable identification data [15]. In addition, the reduced cost of DNA sequencing makes eDNA technology fast, economical, and easy to implement. Thus, the eDNA method offers an accurate, convenient, standardized, and automated solution for routine monitoring of aquatic organisms, including microbes [15,16,17]. Based on this, the eDNA-PFU method was proposed as an enhancement to the PFU method, aiming to monitor the composition and functionality of microbial communities in a more accurate, convenient, and rapid manner [18].

However, the application of eDNA-PFU technology has not yet been fully researched and validated. To successfully implement the eDNA-PFU method for monitoring the aquatic environment, it is essential to obtain various parameters of the eDNA-PFU system and thoroughly evaluate its accuracy and effectiveness in monitoring microbial communities. Therefore, this study represents the first-ever utilization of eDNA-PFU technology in an experiment conducted within a stable and controllable artificial aquatic ecosystem [19,20,21], known as the mesocosm system. Over a period of ten days, we conducted sampling and sequencing of the microbial communities within the PFU while using water environmental samples as controls for comparison. The objectives of this study were: (1) to compare the similarities and distinctions in microbial community composition between PFUs and the surrounding water; (2) to analyze microbial community changes within PFUs and their colonization dynamics; and (3) to explore patterns of interactions within PFUs at the community level using eDNA technology.

## 2. Materials and Methods

### 2.1. Sample Collection and Preparation

The outdoor mesocosm experiment was conducted in October 2021 at Huazhong Agricultural University. The mesocosms comprised three 400 L fiberglass-reinforced plastic tanks with a diameter of 1 m (Figure 1A), and water was obtained from Liangzi Lake in Hubei Province. The PFU blocks had an opening of 200–400 μm and were approximately 5 × 6.5 × 7.5 cm in size. Before use, the PFU blocks were soaked in distilled water for 12 h and then squeezed to remove most of the water [22].

At each test site, several PFU blocks tied together with string were placed in the water and submerged to a depth of 30–50 cm below the water surface (Figure 1A). After 1, 3, 5, 7, and 9 days, one PFU block was randomly removed from each experimental tank, along with samples from the surrounding waters. All PFU blocks and water samples were placed in a cryogenic storage box and transported to the laboratory within three hours. In the laboratory, the PFU blocks were repeatedly rinsed and squeezed with distilled water to obtain the extrudate. The eDNA-PFU and eDNA samples were prepared by filtering the PFU extrudate and water samples with 0.22 μm polycarbonate filter membranes (Millipore Corporation, Billerica, MA, USA), respectively, and stored in an ultra-low temperature freezer at −80 °C.

### 2.2. DNA Extraction, Metagenomic Sequencing and Sequence Assembly

The DNA of the samples was extracted from the filter membranes using the cetyltrimethylammonium bromide (CTAB) method [23,24]. The DNA purity was determined using a nanophotometer (IMPLEN, Westlake Village, CA, USA), and the DNA concentration was measured using the Qubit dsDNA Assay Kit on a Qubit 2.0 fluorometer (Life Technologies, Carlsbad, CA, USA). DNA samples of >1 μg and of qualified quality were then used to construct libraries. The Illumina sequencing connector was ligated to both ends of library DNA using T4 ligase, and all samples were sequenced by Novogene (Beijing, China). The original metagenomic data were processed to obtain high-quality sequencing reads using FastP (v0.11.9) for quality filtering [25]. Filter criteria were an error rate of < 1% and low-quality cut reads of >75 bp in length (-Q 30, -T 0.5). The filtered, high-quality reads were used to extract small subunit (SSU) rDNA reads, which were then used to generate operational taxonomic units (OTUs) representing the entire microbial community present in the sample as follows: First, the SSU sequence was extracted from the metagenomic data using a homology-based method. For this, raw metagenomic reads were aligned to the SILVA database (https://www.arb-silva.de/, accessed on 5 November 2022) [26] and those with homology to the SSU reference database were extracted as candidate sequences. Second, SSU reads from the eDNA and eDNA-PFU samples were assembled separately using Megahit (version 1.1.3) [27]. The assembly parameters were set to default, with the exception that the sequence length had to be greater than 500 base pairs. Additionally, non-redundant Operational Taxonomic Units (OTUs) were constructed for all assembled contigs using cd-hit (version 4.5.4) [28], with the clustering parameter set to 97% identity. The abundance of each OTU was determined and annotated using composition tables of microbial species. Initially, prokaryotic OTUs were annotated using the SILVA database (https://www.arb-silva.de/, accessed on 12 November 2022) [26,29], and eukaryotic OTUs were annotated using the PR2 database (https://github.com/vaulot/pr2_database, accessed on 12 November 2022) [30].

### 2.3. Abundance Assessment of Community Composition and Statistical Analysis

BWA (Burrows-Wheeler-Alignment Tool, version 0.7.17) was used to map the recovered SSU reads to the OTUs, and then for each OTU the relative abundance was determined, the read coverage was calculated using Samtools Bedcov (version 1.3.1) [31], and the relative abundance among samples was calculated using a local R script. Based on the relative abundance of each OTU and species annotation, the Microeco package [32] was used to determine the relative abundance of organisms in each sample at the kingdom, phylum, class, order, family, genus, and species levels. Alpha diversity indices (including the richness index and Shannon-Wiener diversity index) were calculated using the “vegan” package in R (version 4.2.1). Principal coordinate analysis (PCoA) and Permutational Multivariate Analysis of Variance (PERMANOVA) were performed based on the Bray-Curtis distance using the “vegan” package in R software (version 4.2.1) [32]. To group and cluster the fluctuation patterns of different genera normalized to their relative abundance in the PFU, the “Mfuzz” package and the fuzzy c-means method in R software (version 4.2.1) were employed [33,34].

### 2.4. Molecular Ecological Network Analysis nad Cohesion Calculated

The Microbial Ecological Networks (MENs) were constructed based on Spearman correlations at the genera level with a relative abundance ≥ 0.1‰ and occurring in more than 60% of all samples. Genus pairs with correlations of at least −0.6 or 0.6 and a significance level of *p* < 0.01 were retained for MENs analysis. The MENs were visualized in Gephi (version 0.10.1) and Cytoscape (version 3.10.1) software. Within-module connectivity (Zi) and among-module connectivity (Pi) are measures used in network analysis to evaluate the connection strength of a node within a module and between modules, respectively. In this study, these measures were used to classify Genus in the MENs into four subclasses: peripheral nodes (Zi ≤ 2.5 and 0 ≤ Pi ≤ 0.62), connectors (Zi ≤ 2.5 and Pi > 0.62), module hubs (Zi > 2.5 and Pi ≤ 0.62), and network hubs (Zi > 2.5 and Pi > 0.62), following standards laid out in previous studies [35]. Peripheral nodes exhibit low within-module connectivity and low among-module connectivity, while network hubs demonstrate high within-module connectivity and high among-module connectivity [36,37]. To calculate each cohesion metric, we multiplied the relative abundance of each genus in a sample by their associated connectedness values and then summed these products, following the equation laid out in previous studies. Thus, communities with high relative abundances of strongly connected taxa would have a high score of community cohesion. The cohesion index is bounded from −1 to 0 for negative cohesion or from 0 to 1 for positive cohesion [36].

## 3. Results

### 3.1. Overview of Sequencing Data and Microbe Diversity Varied

In this study, 29 samples were obtained, of which 14 were eDNA samples and 15 were eDNA-PFU samples. DNA was extracted from the samples, followed by metagenomic sequencing. A total of 29 of 1013.9 Gb of data were collected, with each sample yielding an average of 34.96 Gb of sequencing data (Appendix A). In total, we recovered 8,817,378 SSU reads from the samples using a homology-based technique, representing an average of 304,048 reads per sample. This reads assembly findings identified 35,879 contigs. After de-redundancy sequences with a similarity of >97%, we obtained 10,837 OTUs for the eDNA-PFU samples and 10,713 for the eDNA samples. These OTUs were assigned to 1065 and 1059 genera, respectively (Appendix A). Rarefaction curves for the eDNA-PFU and eDNA samples were similar, and the sequencing of 0.8 million reads from all samples was sufficient to approach saturation in microbe richness. Rarefaction curves plateaued for eDNA-PFU and eDNA samples, indicating good sampling of the microbe community (Figure 1B).

For eDNA-PFU and eDNA samples, approximately 99% of the OTUs were shared and approximately 1% were unique (Figure 1C). This indicates that the eDNA-PFU method is capable of detecting almost all microbes present in the water environment. The diversity indices show that microbe diversity varied in a temporal manner; both eukaryotic and prokaryotic microbes exhibit stronger variations over colonization time in the eDNA-PFU samples compared to the eDNA samples (Figure 1D,E). In addition, we observed that both the Shannon indices of eukaryotic and bacteria in the eDNA-PFU samples exhibited a pattern of initial increase, followed by a decrease, and then a subsequent increase. This pattern was strikingly different from the eDNA samples, where the Shannon index showed only minor fluctuations. However, whether for bacteria or eukaryotic microbes, the richness indices in both eDNA and eDNA-PFU samples followed a similar pattern, showing small fluctuations in species number over time. This observation suggests that abundance plays a more significant role in driving changes in microbial community diversity over the colonization period. 

### 3.2. The Global Picture of Microbe Community Composition

We analyzed the microbial community composition in eDNA and eDNA-PFU samples based on species composition and abundance information (Figure 2A). In the eDNA-PFU samples, the relative abundance of bacteria ranged from 74% to 87%, and the relative abundance of eukaryotes ranged from 12% to 25%. The relative abundance of archaea was very low, less than 1%. In the eDNA samples, the relative abundance of bacteria was approximately 85%, the relative abundance of eukaryotes was around 14%, and archaea also had the lowest relative abundance, less than 1%. 

For eDNA-PFU samples, the top 10 bacterial taxa with their mean relative abundance are as follows: Proteobacteria (27.92%), Cyanobacteria (19.30%), Bacteroidota (8.95%), Firmicutes (8.60%), Actinobacteriota (3.56%), Elusimicrobiota (2.90%), Verrucomicrobiota (2.69%), Chloroflexi (1.32%), and Acidobacteriota (1.28%) (Figure 2B and Appendix A). The top 10 eukaryote taxa with their mean relative abundance are as follows: Fungi (7.20%), Ochrophyta (2.66%), Ciliophora (1.44%), Cryptophyta (0.98%), Streptophyta (0.79%), Metazoa (0.77%), Chlorophyta (0.73%), Fornicata (0.71%), Euglenozoa (0.52%), and Cercozoa (0.35%) (Figure 2C and Appendix A).For eDNA samples, the top 10 bacterial taxa with their mean relative abundance are as follows: Proteobacteria (25.49%), Cyanobacteria (18.23%), Actinobacteriota (12.64%), Bacteroidota (10.40%), Firmicutes (9.11%), Verrucomicrobiota (3.56%), Elusimicrobiota (2.60%), Chloroflexi (1.11%), and Planctomycetota (1.07%) (Figure 2B and Appendix A). The top 10 eukaryote taxa with their mean relative abundance are as follows: Fungi (4.90%), Ochrophyta (1.95%), Chlorophyta (1.02%), Metazoa (0.81%), Ciliophora (0.65%), Streptophyta (0.55%), Cercozoa (0.43%), Fornicata (0.34%), Dinoflagellata (0.27%), and Cryptophyta (0.21%) (Figure 2C and Appendix A).

Furthermore, principal coordinate analysis (PCoA) and PERMANOVA analysis revealed significant differences in the composition of both bacterial and eukaryotic communities between eDNA-PFU and eDNA samples (Figure 2 D,E). The results showed that PCoA1 and PCoA2 explained 29.0% and 28.7% of the total variation in eukaryotic communities (Figure 2D), and 27.0% and 26.4% of the total variation in bacterial communities (Figure 2E). The PERMANOVA analysis demonstrated that the composition of bacterial and eukaryotic communities between eDNA-PFU and eDNA samples exhibited statistically significant variation (*p* < 0.05). Additionally, we investigated the temporal variation of bacterial and eukaryotic relative abundances over the colonization times (Figure 2F,G). In the eDNA-PFU samples, the relative abundance of bacteria gradually decreased from day 3 to day 7, followed by an increase from day 7 to day 9. In contrast, the relative abundance of eukaryotes exhibited an opposite pattern, gradually increasing from day 3 to day 7 and then decreasing from day 7 to day 9. However, in eDNA samples, both bacterial and eukaryotic relative abundances remained relatively stable, with minimal changes over time. This observation further supports the results of a deterministic colonization process exhibited by microbes in the PFU, consistent with the results from diversity analyses. 

### 3.3. Colonizational Dynamic Patterns of Microbial Community

To investigate the temporal dynamics of microbial colonization in mesocosm PFU systems, we analyzed trends in the relative abundance of different genera over time. By normalizing the trends of species and then clustering them, we were able to divide them into four distinct clusters, each exhibiting a unique temporal-dynamic colonization pattern. In cluster 1, the species relative abundance began to decline on day 1, reached its lowest point on day 3 and remained relatively stable over the subsequent time. In cluster 2, the relative abundance of the species peaked on day 3, declined on day 7, and then rose or equilibrated again. In cluster 3, the species relative abundance peaked on day 5 and 9, respectively. In cluster 4, the species relative abundance peaked on day 7, and then declined (Figure 3A).

In addition, we conducted a statistical analysis of the distribution of various taxa in the four clusters (Figure 3B,C). We observed that most eukaryotic genera were predominantly distributed in clusters 3 and 4, while bacterial genera were mainly distributed in clusters 1 and 2. Notably, the genus number of Fungi, Ciliophora, Fornicata, and Chlorophyta in cluster 4 was higher compared to other clusters. However, the genus number of Metazoa showed small variations across all four clusters. As for bacteria, Proteobacteria, Cyanobacteria, Bacteroidota, Firmicutes, and Actinobacteriota exhibited a higher genus number in clusters 1 and 2 compared to other clusters. Moreover, Cyanobacteria displayed a distribution pattern similar to that of eukaryotes, mainly concentrated in clusters 3 and 4.

In summary, our findings suggest that protists (such as Ciliophora, Cyanobacteria, and Chlorophyta) and fungi exhibited a gradual increase in abundance over colonization time in eDNA-PFU samples. On the other hand, most bacteria (excluding Cyanobacteria) mainly showed a decreasing trend.

### 3.4. Molecular Ecological Network of Microbial Communities

MENs analysis was performed to explore potential relationships between microbial communities present in the PFU and water. The microbial MENs under eDNA-PFU and eDNA samples followed different patterns on the basis of 11 network topological parameters (Table 1). We also constructed random networks, and the results show that the average path length, average clustering coefficient, and modularity of the empirical networks are higher than the values of their respective random networks (Table 1 and Appendix A). Although the numbers of genera used for network construction were on average 23% fewer under eDNA than under eDNA-PFU, the resulting networks without isolated nodes were on average 14% larger in size (total nodes) under eDNA than under eDNA-PFU, which indicates higher network modularity in eDNA-PFU and higher network connectivity in eDNA. The MENs of the eDNA-PFU sample, as shown in Figure 4A, show that the predominant phylum microbes were Proteobacteria, Ochrophyta, Bacteroidota, Ciliophora, Actinobacteriota, Fungi, Metazoa, Chlorophyta, and Cyanobacteria (Appendix A). The MENs of the eDNA sample, as shown in Figure 4B, show that the predominant phylum of microbes was Proteobacteria, Bacteroidota, Actinobacteriota, Chlorophyta, Cyanobacteria, Ochrophyta, Fungi, Metazoa, Verrucomicrobiota, Ciliophora, Firmicutes, Dinoflagellata, Cercozoa, Desulfobacterota, Streptophyta, and Cryptophyta (Appendix A). 

To elucidate functional variations within microbial MENs, we compared the connectivity of MENs between eDNA-PFU and eDNA samples (Figure 4C,D). The findings reveal that the positive connectivity of Chlorophyta, Ciliophora, Cryptophyta, Fungi, Metazoa, and Bacteroidota is higher in eDNA-PFU compared to eDNA samples. However, positive connectivity was higher in the eDNA samples for Ochrophyta, Proteobacteria, and Actinobacteriota, while negative connectivity was higher for all taxa except Acidobacteriota and Bacteroidota. 

Furthermore, to elucidate the variation patterns of MENs over colonization time, we calculated the network cohesion value and analyzed the relationship between cohesion values and Bray-Curtis. We used multiple regression to model compositional turnover (Bray-Curtis dissimilarity between colonization time (day 1)) as a function of community cohesion at the initial time point. That is, Bray-Curis dissimilarity was the dependent variable, whereas positive and negative cohesion were the independent variables. Cohesion values (both positive and negative) were calculated at the first time point for each sample pair. The results showed that in the eDNA-PFU samples, there was a significant positive correlation between Bray-Curtis dissimilarity and both positive and negative cohesion, with R values of 0.413 and 0.350, respectively. However, in the eDNA samples, there was a significant negative correlation between Bray-Curtis dissimilarity and both positive and negative cohesion, with R values of 0.97 and 0.66 (Figure 5A and Appendix A). This indicates that cohesion indices are closely related to microbial community dynamics in MENs. 

In addition, we further explored the variation of cohesion values with time of colonization, as shown in Figure 5B. The results indicate that the trends of cohesion value (positive cohesion and negative cohesion) in eDNA and eDNA-PFU samples are completely opposite. Specifically, there is a decreasing trend in the positive cohesion index in eDNA samples, while it exhibits an increasing trend in eDNA-PFU samples, although this trend is not statistically significant. However, the negative cohesion index in eDNA-PFU shows a significant decreasing trend. These findings suggest that the network structure of eDNA-PFU MENs experiences a decrease in negative cohesion with colonization time, resulting in a higher degree of modularity in their networks.

The altered network structure could lead to changes in the roles of individual members within the network. On the basis of their within-module connectivity (Zi) and among-module connectivity (Pi), a total of 8 module hubs (nodes highly connected to other members in a module), 6 connectors (nodes linking different modules), and 306 peripheral hubs were detected across all the MENs in eDNA-PFU samples. However, a total of 0 module hubs, 2 connectors, and 253 peripheral hubs were detected across all the MENs in the eDNA samples. The module hubs and module hubs could be regarded as keystone nodes (Methods) that play key roles in shaping network structure (Figure 5C,D). We observed that the module hubs in eDNA-PFU samples were primarily represented by Ciliophora, Fungi, and Ochrophyta. These findings suggest that these eukaryotic taxa play significant roles in the Microbial Ecological Networks (MENs) of PFU.

## 4. Discussion

### 4.1. Efficiency of the eDNA-PFU Method in Mesocosm Experiments for Bioassessments

Previous investigations on the colonization of microbial communities using the PFU method have primarily focused on natural ecosystems, such as rivers, lakes, and marine environments [10,12,38,39,40]. The results of these studies indicate that PFU, as an effective bio-detection method, has been widely utilized for the study of microbial detection and colonization patterns in water environments. However, conventional morphology-based PFU methods used in investigations of microbial colonization patterns have some notable limitations. Firstly, these studies mainly focus on protists or protozoans [9,12,41,42], often neglecting other microbial groups such as bacteria and fungi, which are known to be integral components of aquatic ecosystems [43,44,45]. Therefore, it is essential to also consider these taxa when studying microbial community patterns. Secondly, many studies are conducted in relatively open natural water bodies, where factors such as water temperature and flow rate may introduce significant interference and variability in the determination of microbial colonization parameters [46,47].

Therefore, in this study, we investigated the microbial communities in controlled and relatively stable artificial water ecosystems (mesocosms) using metagenomic sequencing technology. We compared the eDNA-PFU method with the eDNA method for biological detection in water environments. As shown in the results section, both eDNA-PFU and eDNA methods are effective in capturing microbial taxa, including eukaryotes and prokaryotes, present in aquatic environments. Notably, eDNA-PFU exhibited remarkable sensitivity by detecting 99.95% of the species identified by the eDNA method. And eDNA-PFU displayed significantly higher microbial diversity, especially in terms of abundance. Furthermore, a noteworthy observation was the reduced fluctuation in the number of microbial species detected by eDNA-PFU in comparison to the eDNA method. This may be attributed to the eDNA-PFU method immersing the PFU sponge block in the water for at least 24 h during sample collection, leading to the enrichment of rare or low-abundance species in the water environment [18,48,49]. Consequently, this method detects a higher number of microbial species while reducing the random error in sample collection, resulting in lower fluctuations in the eDNA-PFU method. Overall, these findings demonstrate that the eDNA-PFU method represents an effective microbial detection technique. 

### 4.2. Colonization Patterns of Microbial Community in Mesocosm Experiments

Exploring microbial colonization patterns is crucial for comprehending the dynamics and potential interactions during microbial community assembly [39,40]. In this study, we observed distinct colonization of microbial communities in the PFU system over time. Notably, an analysis of changes in diversity revealed that the diversity index of the microbial community in the PFU system started to decrease on the third day, indicating that the colonization of microbes in the PFU likely reached equilibrium at this point. Traditionally, the process of PFU colonization to equilibrium has been considered lengthy, taking around 2–5 weeks for static water and approximately 3–7 days for running water [6,8,50]. However, with the eDNA-PFU system, we found that PFUs reached equilibrium much faster than the time estimated by conventional methods. This discrepancy may be attributed to the fact that identification by traditional methods might not cover the entire microbial community.

Additionally, our investigation into relative abundance revealed intriguing temporal trends, showing notable differences in the colonization process between eukaryotes and bacteria. This observation suggests potential interconnections between these two groups during colonization [50]. We delved further into the analysis by examining the relative abundance of different taxa at the phylum level of bacteria and eukaryotes over time. The results revealed significant colonization patterns for Fungi, Ochrophyta, Ciliophora, and Cercozoa, with a notable increase in relative abundance starting from day 5, consistent with the findings of Xu’s research [51]. As for bacteria, Proteobacteria showed the most noticeable decrease in relative abundance, while Cyanobacteria exhibited a significant increase from day 5. However, these taxa did not exhibit substantial relative abundance changes in the eDNA samples, showing only minor fluctuations. This further confirms the presence of microbial colonization processes in the PFU system. 

We classified the microbial community colonization patterns in the PFU system into four clusters. Eukaryotes showed a higher concentration in Cluster 3 and Cluster 4, both indicative of slow colonization patterns. On the other hand, prokaryotes were predominantly distributed in Cluster 1 and Cluster 2, representing faster colonization patterns. And different taxa exhibit varied colonization patterns. For instance, fungi primarily conformed to Cluster 4, while Cercozoa appeared exclusively in Cluster 3. Moreover, Firmicutes and Proteobacteria were predominantly associated with Clusters 1 and 2, while Cyanobacteria exhibited dominance in Clusters 3 and 4. These findings reveal significant differences in the colonization rates of various microbial communities within the PFU system. These disparities could be attributed to factors such as nutrient availability, individual size, and other ecological dynamics [39,40]. Notably, our results align with those obtained through conventional methods, reinforcing the validity and consistency of the observed colonization patterns.

Overall, these findings provide valuable insights into the intricate colonization dynamics of microbial communities in the PFU system, enhancing our understanding of ecological interactions in aquatic environments. The observed fast-reaching equilibrium and the notable differences in colonization patterns underscore the significance of employing advanced methods such as eDNA-PFU for gaining comprehensive insights into microbial community assembly and dynamics. By utilizing such innovative approaches, we can further unravel the complexities of microbial interactions and their roles in shaping aquatic ecosystems. 

### 4.3. Molecular Ecological Network among Microbial Communities

In ecosystems, various species interact through the exchange of materials, energy, and information, creating intricate connections [52,53]. Molecular Ecological Networks (MENs) not only unveil intricate interspecies relationships but also reveal the dynamic changes in microbial community functional structures and related ecological processes through network dynamics [37,54].

The MENs analysis revealed that the eDNA-PFU samples exhibited higher modularity compared to the eDNA samples. We quantified community complexity using the cohesion index, where positive cohesion reflects cooperative behaviors among samples and negative cohesion indicates the extent of competitive behaviors among community members [36]. Remarkably, we observed a significant decreasing trend in the negative cohesion index of eDNA-PFU with colonization time. This finding indeed explains the main reason for the increasing modularity of the network in the PFU and the enhanced connections within the network modules [55].

Moreover, our analysis based on within-module connectivity (Zi) and among-module connectivity (Pi) revealed that module hubs in PFU were primarily composed of Proteobacteria, Ochrophyta, Ciliophora, and Fungi. This finding aligns well with our earlier analysis of the colonization pattern, providing further evidence of the distinct and specialized microbial communities in PFU.

These findings provide us with valuable insights into understanding the interactions among microbials within ecosystems. They contribute to a deeper comprehension of ecosystem complexity and the crucial roles played by microbials within them. In the future, it is imperative to continue using eDNA-PFU technology in experiments within natural aquatic ecosystems. This will enable us to further investigate the intricate relationships, characteristics, and impact on ecosystem functionality within PFU microbial communities. 

## 5. Conclusions

Our study provides valuable insights into the colonization dynamics and network interactions of microbial communities in mesocosm ecosystems using eDNA-PFU technology. The eDNA-PFU method proves to be a powerful and efficient tool for assessing microbial diversity in water environments, capturing nearly all microbial taxa present. The colonization patterns observed in the PFU system indicate distinct differences in colonization trends between eukaryotes and bacteria, with Fungi, Ochrophyta, Ciliophora, and Cercozoa displaying notable colonization patterns. These findings shed light on the complexity of microbial community assembly and dynamics in aquatic ecosystems. Furthermore, the analysis of molecular ecological networks revealed that eDNA-PFU samples exhibited higher modularity, emphasizing the presence of distinct and specialized microbial communities within the PFU system. The identification of key module hubs comprising Proteobacteria, Ochrophyta, Ciliophora, and Fungi further highlights their crucial roles in shaping the MENs of PFU. 

In summary, this study demonstrates the potential of eDNA-PFU technology for gaining comprehensive insights into microbial ecology in water environments. The understanding of colonization dynamics and network interactions has laid a solid foundation for further unraveling the complexity of microbial communities within PFUs. This endeavor not only aids in developing reliable methods for assessing microbial diversity and environmental conditions in aquatic ecosystems but also offers invaluable insights for the better preservation and management of these ecosystems. 

## Figures and Tables

**Figure 1 microorganisms-11-02498-f001:**
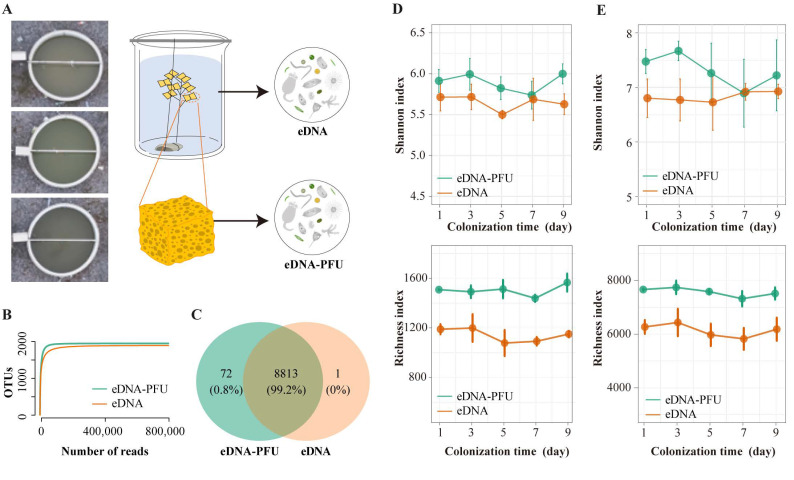
Mesocosm experiment strategy for eDNA-PFU and eDNA samples and basic statistical analyses. (**A**) Map illustrating the design of the mesocosm experiment; (**B**) Rarefaction curves for microbial communities in eDNA-PFU and eDNA samples; (**C**) Venn diagram showing the number of shared and specific OTUs for eDNA-PFU and eDNA samples; (**D**) Trends of alpha diversity of eukaryotic (**D**) and bacteria (**E**) microbial communities in eDNA-PFU and eDNA samples.

**Figure 2 microorganisms-11-02498-f002:**
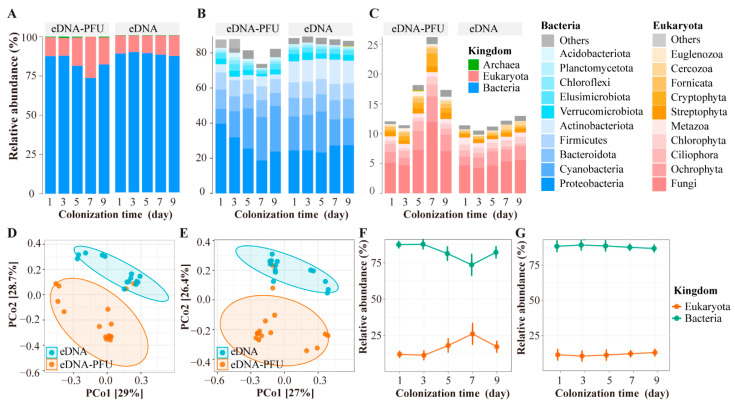
Species composition and abundance. (**A**) Relative abundance of OTUs at each sampling colonization day in the different microbial communities (**B**,**C**). (**D**,**E**) Analysis of PCOA in the eukaryotic (**D**) and bacterial communities (**E**). (**F**,**G**) Trends in the relative abundance of eukaryotes in PFU (**F**) and water (**G**).

**Figure 3 microorganisms-11-02498-f003:**
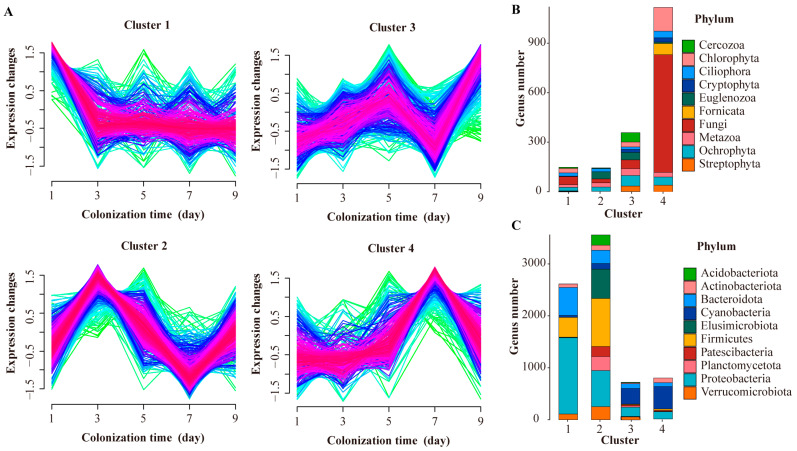
Temporal trends in the abundance of microbial communities. (**A**) Fuzzy c-means clustering identified trends in species abundance at the genus level in PFU samples, the color from green to red indicates an increase in density of the change trendline. Number of genera per phylum for each cluster in PFU samples in the eukaryotic (**B**) and bacteria communities (**C**).

**Figure 4 microorganisms-11-02498-f004:**
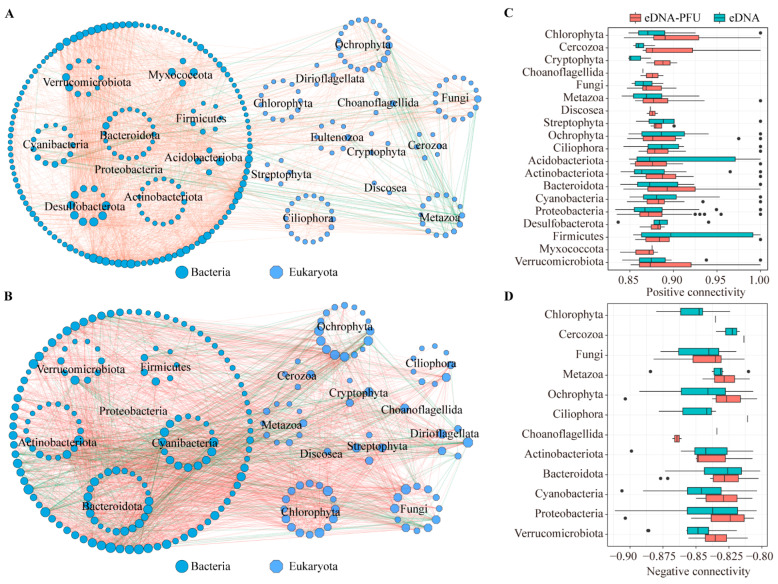
Molecular ecological network analysis. Molecular ecological network of microbial communities in the eDNA-PFU (**A**) and eDNA (**B**) samples. The positive (**C**) and negative (**D**) connectivity of the network in the eDNA-PFU and eDNA samples.

**Figure 5 microorganisms-11-02498-f005:**
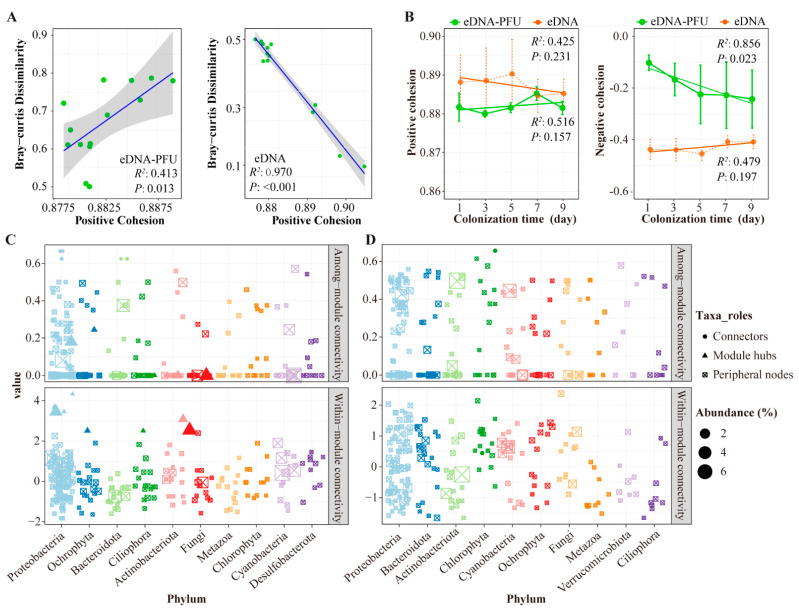
Molecular ecological network analysis. (**A**) Relationship between cohesion and compositional turnover (Bray-Curtis dissimilarity) in microbial communities. (**B**) Changes in cohesion (positive cohesion and the absolute value of negative cohesion) of MENs across colonization times in PFU and water samples (**C**,**D**) Scatter plot of Zi and Pi of MENS in PFU (**C**) and water (**D**) samples. The threshold values of Zi and Pi for categorizing genera were 2.5 and 0.62, respectively.

**Table 1 microorganisms-11-02498-t001:** Topological parameters of MENs for eDNA-PFU and eDNA samples.

Characteristic	eDNA-PFU		eDNA	
	Empirical Network	Random Network	Empirical Network	Random Network
Nodes	332	332	255	255
Edges	2242	2242	2550	2550
Average degree	13.51	13.50 ± 0.001	20	9.05 ± 1.18
Average path length	4.00	2.52 ± 0.002	2.64	2.11 ± 0.0026
Network diameter	12	4 ± <0.001	9	3.01 ± 0.1
Clustering coefficient	0.62	0.041 ± 0.002	0.58	0.078 ± 0.001
Centralization degree	0.12	0.035 ± 0.005	0.133	0.0517 ± 0.006
Heterogeneity	1.04	1.04	0.75	0.75
Connectance	0.004	0.004	0.079	0.079
Centralization betweenness	0.12	0.01 ± 0.003	0.045	0.007 ± 0.001
Centralization closeness	0.16	0.09 ± 0.013	1.31	0.092 ± 0.001

## Data Availability

The data presented in this study are deposited in the National Genomics Data Center repository, accession number PRJCA018872.

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
