# Peer review of "Microbial Community Colonization Process Unveiled through eDNA-PFU Technology in Mesocosm Ecosystems"

_microorganisms, 2023, doi:10.3390/microorganisms11102498_

Round 1
Reviewer 1 Report
In this manuscript the authors compared two environmental monitoring techniques; the polyurethane foam unit (PFU) method, and the improved eDNA-PFU method. Overall, the article is well written and easy to read. The experimental procedures are technically sound, extensively described and discussed. The results were also presented in clear figures, appropriately discussed and compared to current literature. The only recommendation for follow-up studies is to test the tools in a flow system and not in stationary systems. This would similate a more realistic environmental scenario.
Author Response
Dear Editor,
Thank you for the opportunity to resubmit our revised manuscript for further review by Microorganisms. We are very grateful for the valuable comments of the reviewers. We have addressed the reviewers’ concerns in the point-by-point response and revised the manuscript accordingly, with major textual changes highlighted with tracked change model.
Reviewer #1
In this manuscript the authors compared two environmental monitoring techniques; the polyurethane foam unit (PFU) method, and the improved eDNA-PFU method. Overall, the article is well written and easy to read. The experimental procedures are technically sound, extensively described and discussed. The results were also presented in clear figures, appropriately discussed and compared to current literature. The only recommendation for follow-up studies is to test the tools in a flow system and not in stationary systems. This would similate a more realistic environmental scenario.
Response: we thank the reviewer for the overall positive comments about this manuscript. We have incorporated the prospects of conducting experiments in natural aquatic environments into the discussion section (see Discussion 4.3). Additionally, we have initiated work in this direction.
Reviewer 2 Report
I would think a bit of attention to the nature of the surfaces on which the biofilms grow might add to understanding the relevance of the work.
I am into water quality improvements and microbial role in treatments of water and have difficulties to understand the practical implementation of the method in environmental monitoring water quality. The manuscript is great, but I am an old environmental scientist.
Author Response
Dear Editor,
Thank you for the opportunity to resubmit our revised manuscript for further review by Microorganisms. We are very grateful for the valuable comments of the reviewers. We have addressed the reviewers’ concerns in the point-by-point response and revised the manuscript accordingly, with major textual changes highlighted with tracked change model.
Reviewer #2
I would think a bit of attention to the nature of the surfaces on which the biofilms grow might add to understanding the relevance of the work.
I am into water quality improvements and microbial role in treatments of water and have difficulties to understand the practical implementation of the method in environmental monitoring water quality. The manuscript is great, but I am an old environmental scientist.
Response: Thank you for your suggestion. The eDNA-PFU method represents an improvement upon the conventional PFU method, aiming to address issues associated with the high level of expertise and subjectivity required by conventional PFU techniques. Our research aims to optimize eDNA-PFU for environmental monitoring. Although research on eDNA-PFU is limited, we are gaining valuable experience.
Our study shows that PFU analysis provides more comprehensive microbial information compared to direct water sampling. Microorganisms can colonize PFUs, reducing randomness. Studying microbial community dynamics in PFUs improves specific group detection. PFU microbial communities are unique and interact closely, enabling community-level monitoring.
These findings serve as a vital theoretical and practical foundation for further refinement and enhancement of the eDNA-PFU method, thereby better meeting the demands of environmental monitoring. Currently, we have initiated research experiments in natural aquatic systems and will also place increased emphasis on investigating the nature of surfaces where biofilm growth occurs. We remain committed to accumulating more data and experience to continually improve this method. Once again, we appreciate your valuable suggestions and support.

Reviewer 3 Report
The article entitled “Microbial community colonization process unveiled through eDNA-PFU technology in mesocosm ecosystems” submitted to Microorgnisms focuses on the use of the eDNA-PFU method to study the colonisation process of microorganisms in PFUs. To achieve this, eDNA-PFU experiments were carried out in a stable and controlled mesocosm system and control experiments were performed using eDNA methods. The paper is written to a high standard and includes all elements of structure. The methodology of the work is well chosen and described in detail. The manuscript only needs minor revision in some sections. Details are provided below:
Abstract: The abstract is too long and needs to be shortened
Introduction: Please give more justification for the importance of the issue addressed and indicate the novelty of the paper
The aim of the work is general and requires rewriting.
Discussion
In my opinion, figures and tables should not be cited in the discussion and should be removed.
The results and their implications should be discussed in a broader context. The limitations of the methods used should also be highlighted and discussed.
Author Response
Dear Editor,
Thank you for the opportunity to resubmit our revised manuscript for further review by Microorganisms. We are very grateful for the valuable comments of the reviewers. We have addressed the reviewers’ concerns in the point-by-point response and revised the manuscript accordingly, with major textual changes highlighted with tracked change model.
Reviewer #3
Abstract:
- The abstract is too long and needs to be shortened
Response: revised.
Introduction:
- Please give more justification for the importance of the issue addressed and indicate the novelty of the paper
Response: revised.
- The aim of the work is general and requires rewriting.
Response: revised.
Discussion:
- In my opinion, figures and tables should not be cited in the discussion and should be removed.
Response: revised.
- The results and their implications should be discussed in a broader context. The limitations of the methods used should also be highlighted and discussed.
Response: We have added this kind of discussion (see Discussion 4.3).